# Unveiling the Chemistry of Higher-Order Cycloaddition Reactions within the Molecular Electron Density Theory

**Luis R. Domingo [1],\*, Mar Ríos-Gutiérrez [1] and Patricia Pérez [2]**

[1] Department of Organic Chemistry, University of Valencia, Dr. Moliner 50, Burjassot, 46100 Valencia, Spain; m.mar.rios@uv.es

[2] Departamento de Ciencias Químicas, Centro de Química Teórica & Computacional, Facultad de Ciencias Exactas, Universidad Andrés Bello, Santiago 8370146, Chile; p.perez@unab.cl

\* Correspondence: luisrdomingo@gmail.com

**Abstract:** The higher-order cycloaddition (HOCA) reaction of tropone with cyclopentadiene (Cp) has been studied within the Molecular Electron Density Theory. The Electron Localization Function (ELF) analysis of the electronic structure of tropone and Cp characterizes the structural behaviors of the two conjugated unsaturated systems, while the conceptual DFT reactivity indices classify tropone as a strong electrophile and Cp as a strong nucleophile participating in polar cycloaddition reactions of reverse electron density flux. Eight competitive reaction paths have been characterized for this cycloaddition reaction. The most favorable one allowing the formation of the formal *out* [6 + 4] cycloadduct has an activation enthalpy of 16.2 kcal·mol$^{-1}$, and the reaction is exothermic by −21.4 kcal·mol$^{-1}$. This HOCA reaction, which takes place through a non-concerted *two-stage one-step* mechanism, presents high stereo-, *pseudocyclic-* and regioselectivities, explaining the exclusive formation of the experimental [6 + 4] cycloadduct. While the most favorable nucleophilic attack of Cp on most electrophilic C2 positions of tropone accounts for regioselectivities, the favorable electrostatic interactions present between the Cp framework and the negatively charged O8 oxygen of tropone account for the stereo- and *pseudocyclic* selectivities. Despite the symmetry of the two reagents, this HOCA reaction takes place via a highly asynchronous transition state structure as a consequence of the most favorable two-center interactions taking place between the electrophilic C2 center of tropone and the nucleophilic C9 center of Cp.

**Keywords:** higher-order cycloaddition reactions; molecular electron density theory; molecular mechanism; tropone; selectivity

## 1. Introduction

Cycloaddition reactions are among the most important reactions in Organic Chemistry, allowing the construction of any cyclic compound with highly stereo- and regioisomeric outcomes [1,2]. The Diels–Alder or [4 + 2] cycloaddition reaction between a diene and an ethylene derivative is the most common [3]. When a conjugated triene such as tropone **1**, which is a carbonyl derivative of cycloheptatriene **2**, is used to build larger cyclic compounds, cycloadditions such as [4 + 2], [4 + 3], [4 + 4], [6 + 2], and even [8 + 2] become possible. The formation of different constitutional isomeric [m + n] cycloadducts (CAs) is achievable in these higher-order cycloaddition (HOCA) reactions involving three or more double bonds that allow for the construction of medium-size rings [4] with diverse regio- and stereoisomeric possibilities.

In 1996, Ito et al. reported the reaction of tropone **1** with cyclopentadiene (Cp) **3** [5]. Thus, the heating of tropone **1** with Cp **3** in benzene at 80 °C yielded the 1:1 CA **4** (see Scheme 1). Three structures were initially proposed: the [6 + 4] CAs **4** and **5**, and the [2 + 4] CA **6**. An analysis of the NMR spectra of the reaction product discarded structures **5** and **6**.

**Scheme 1.** HOCA reaction of tropone **1** with Cp **3**.

Concurrently, Cookson et al. reported the reaction of tropone **1** with Cp **3** in the dark for three days at room temperature, obtaining the [6 + 4] CA **4** [6]. By heating CA **4** in xylene reflux, this compound was dissociated into the original components. Again, the analysis of the NMR spectra of the reaction product confirmed the structure of CA **4**. The formation of the [6 + 4] CA **4** involves the participation of the symmetric C2 and C7 carbons of tropone **1**.

The reaction of tropone **1** with dimethylfulvene **7** forms three different CAs, showing the complexity of HOCA reactions (see Scheme 2) [7,8]. While the 1:1 CAs **8** and **9** are a pair of constitutional isomers, the major 2:1 product **10** was obtained via a domino reaction involving a second addition of tropone **1** to CA **9**. Interestingly, the formation of CAs **8** and **9** involves the C2 and C5 carbons of tropone **1**, while the formation of the domino CA **10** involves the C2 and C7 carbons of a second molecule of tropone **1**. Thus, HOCA reactions can occur through different reaction paths involving different molecular mechanisms.

**Scheme 2.** Reaction of tropone **1** with dimethylfulvene **7**.

HOCA reactions of tropone **1** have recently been theoretically studied [4,7–14]. The Woodward–Hoffmann's symmetry rules [15] (W-H SR) and the Frontier Molecular Orbital [16] (FMO) theory have been widely used to explain the course of HOCA reactions [9,13,14]. In 2016, Domingo proposed the Molecular Electron Density Theory [17] (MEDT) to study organic chemical reactivity. MEDT proposes that changes in electron density in an organic reaction, and not molecular orbital (MO) interactions such as the FMO theory proposed [16], are responsible for the organic chemical reactivity. Within MEDT, a series of quantum chemical tools developed at the end of the last century, such as conceptual density functional theory (CDFT) indices [18,19], atom-in-molecules [20] (AIM), electron localization function [21] (ELF), and non-covalent interactions [22] (NCI), which permit the analysis of the molecular electron density, was used to study the structure and reactivity in organic chemistry.

Recently, the HOCA reactions of tropone **1** with the nucleophilic ethylene **11**, in the absence and the presence of Lewis acids (LAs), have been studied within MEDT [23] (see Scheme 3). The strong electrophilic character of tropone **1**, enhanced by the presence of LAs, allows its participation in polar cycloaddition reactions of reverse electron-density flux [24] (REDF) towards nucleophilic ethylenes [23]. These polar HOCA reactions take place via non-concerted *two-stage one-step* [25] or two-step mechanisms, yielding only one 1:1 CA with a total regio- and *pseudocyclic* selectivity [26]. LAs not only accelerate the reaction and make it completely regioselective but also determine the *pseudocyclic* selectivity exclusively

yielding [4 + 2] or [8 + 2] CAs [26], which was found to depend on a series of weak attractive/repulsive intramolecular electronic interactions present at the corresponding diastereoisomeric TSs [23].

**Scheme 3.** Cycloaddition reactions of tropone **1** with the nucleophilic ethylene **11**.

In 2002, Yamabe et al. studied the asymmetry in symmetric cycloaddition reactions using the W-H SR and the FMO theory [9]. For the HOCA reaction of tropone **1** with Cp **3**, the authors suggested that "*in spite of the symmetry of the reagents, the reaction proceeds via asymmetric transition state structure (TS) that does not follow the W-H SR*".

Very recently, Houk et al. reported a ωB97X-D/def2-TZVP computational study of the HOCA reaction of tropone **1** with Cp **3** (see Scheme 4) [27]. These authors suggested that "*these cycloadditions involve ambimodal TSs resulting in a web of products by pericyclic interconversion pathways*" [27]. Four TSs and four CAs were found for this reaction [27]. The *exo* **TS-A** was found at 3.3 kcal·mol$^{-1}$ (DLPNO-CCSD(T)/cc-pVQZ//ωB97X-D/def2-TZVP) and was more favorable in Gibbs-free energy than the *endo* **TS-B**, suggesting that both TSs are subject to "secondary orbital interactions" (SOI) [28]. These authors suggested that the *endo* [6 + 2] CA **4** could be converted in the non-observed formal [8 + 2] CA **15** via a Claisen **TS-C**, while the *exo* [6 + 2] CA **5** could be converted in the non-observed formal [4 + 2] CA **16** via a Cope **TS-D**. They concluded that "*The observed product forms via an ambimodal TS that can lead to [6 + 4] and [8 + 2] adducts, but forms **4** for dynamic and thermodynamic reasons. The unobserved endo **TS-B** is similarly ambimodal and complex. These results lead us to hypothesize that all endo-[6 + 4] cycloadditions are ambimodal*" [27].

**Scheme 4.** Reaction paths characterized by Houk in the HOCA reaction of tropone **1** with Cp **3**.

Many theoretical studies based on the Bonding Evolution Theory (BET) carried out in this century have proven that the "*pericyclic mechanism*", proposed in 1969 by Woodward and Hoffmann for the Diels–Alder reaction [15], does not exist as the bonding changes along one-step organic reaction paths are non-concerted but sequential [29–31]. In addition, as the competitive reaction paths allowing the formation of constitutional isomers do not have a "*pericyclic mechanism*", the term "*periselectivity*" [7,32] proposed by Houk in 1970 for the formation of constitutional isomers should be rejected [33].

The concept of *pseudocyclic* TSs was recently introduced to characterize the TSs in which all nuclei participating in a chemical reaction show a cyclic rearrangement, but not all neighboring atoms need to be bonded [33]. The name *pseudocyclic* reaction is used to categorize these types of organic reactions [33]. The *pseudocyclic* selectivity concept was

recently proposed, providing a more precise definition of the selectivity in the formation of constitutional isomers resulting from competitive stereoisomeric *pseudocyclic* TSs [33]. This concept frequently appears in HOCA reactions [23,34].

Our interest in cycloaddition reactions, and the exhibited complexity of the HOCA reaction of tropone **1** with nucleophilic species [23,34], prompted us to perform an MEDT study of the HOCA reaction of tropone **1** with Cp **3**, independently studied by Ito [5] and Cookson [6], which experimentally yielded only the formal [6 + 4] CA **4**.

## 2. Computational Methods

DFT calculations were performed using the $\omega$B97X-D functional [35], which includes long-range exchange (denoted by X) corrections as well as the semiclassical London-dispersion correction (indicated by suffix D). The standard 6-311G (d,p) basis set was used [36], which includes d-type polarization for second row elements and p-type polarization functions for hydrogen atoms. The Berny method was used in optimizations [37,38]. Only one imaginary frequency characterized all studied TSs. The intrinsic reaction coordinate (IRC) paths [39] were carried out to find the unique connection given between the TSs and the minimum stationary points [40,41]. Solvent effects of benzene were considered by a full optimization of the gas-phase structures at the same computational level using the polarizable continuum model [42,43] (PCM) in the framework of the self-consistent reaction field [44–46] (SCRF). Values of $\omega$B97X-D/6-311G (d,p) enthalpies, entropies, and Gibbs-free energies in benzene were calculated with standard statistical thermodynamics at 80 °C and 1 atm [36], by PCM frequency calculations at the solvent optimized structures.

The global electron density transfer [47] (GEDT) values were estimated by a Natural Population Analysis (NPA) [48,49] by using the equation GEDT(f) = $\Sigma_{q\epsilon f}\, q$, where $q$ denotes the atoms of a framework (f) at the TSs. CDFT reactivity indices [18,19], computed at the B3LYP/6-31G (d) level, were calculated using the equations provided in reference [19]. All calculations were carried out using the Gaussian 16 suite of programs [50].

The topology of the ELF [21] of the $\omega$B97XD/6-311G (d,p) monodeterminantal wavefunctions was carried out using the TopMod [51] package with a cubical grid of step size of 0.1 Bohr. The GaussView program [52] was used to visualize the molecular geometries of all the systems and the position of the ELF basin attractors.

## 3. Results and Discussion

### 3.1. Analysis of the Ground State Electronic Structures of the Reagents

ELF [21,53] permits a quantitative characterization of the electron density distribution in a molecule, establishing a correlation between its electronic structure and its reactivity. Consequently, an ELF topological analysis of the electronic structure of tropone **1** and Cp **3** was first performed. The ELF valence basins, ELF basin attractor positions with the most relevant valence basin populations, and ELF-based Lewis-like structures of **1** and **3** are shown in Figure 1.

The ELF analysis of tropone **1** shows the presence of one V(C1,C2) disynaptic basin, integrating 2.32 e; two disynaptic basins, V(C2,C3) and V′(C2,C3), integrating a total of 3.22 e; one V(C3,C4) disynaptic basin, integrating 2.34 e; two disynaptic basins, V(C4,C5) and V′(C4,C5), integrating a total of 3.17 e; one V(C1,O8) disynaptic basin, integrating 2.27 e; and two monosynaptic basins, V(O8) and V′(O8), integrating 2.67 e for each one. Note that the C1–C7, C7–C6, and C6–C5 bonding regions of tropone **1** are symmetrical to the C1–C2, C2–C3, and C3–C4 ones.

While the V(C1,C2) and V(C3,C4) disynaptic basins are associated with Ci–Cj single bonds, the pairs of V(C2,C3), V′(C2,C3), and V(C4,C5), and V′(C4,C5) disynaptic basins are associated with depopulated Ci–Cj double bonds. These behaviors result from the cyclic conjugation of the three Ci–Cj double bonds with the carbonyl C1 carbon in tropone **1**. The V(C1,O8) disynaptic basin is associated with a significantly depopulated carbonyl C–O double bond, while the two V(O8) and V′(O8) monosynaptic basins are associated with the non-bonding electron density present at the O8 oxygen. These behaviors show the

strong polarization of the electron density of the carbonyl C1–O8 group towards the highly electronegative O8 oxygen.

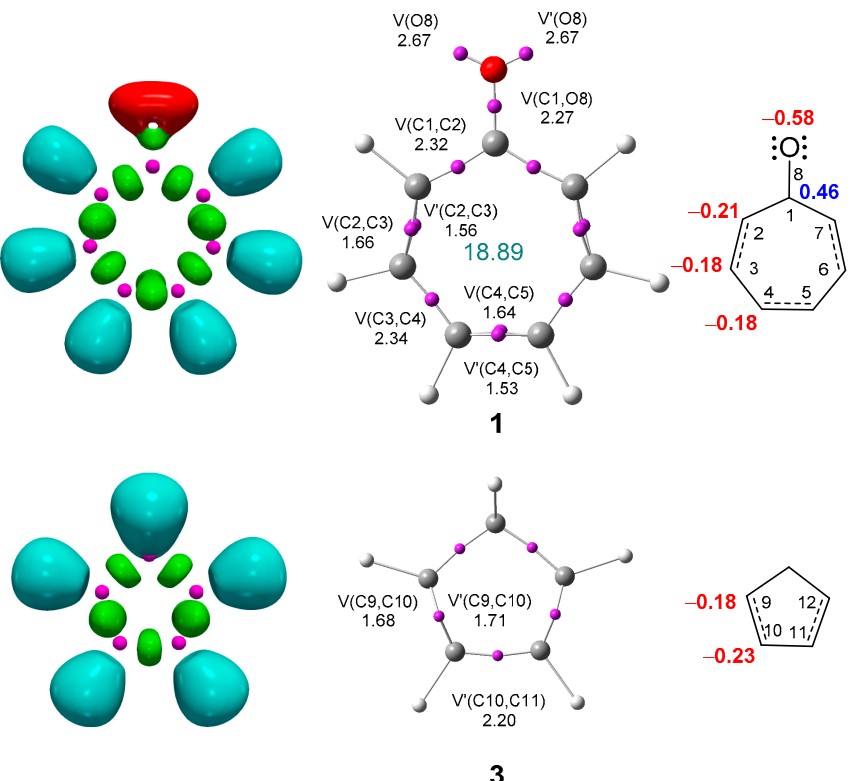

**Figure 1.** ELF localization domains represented at an isosurface value of ELF = 0.75, ELF basin attractor positions together with the most relevant valence basin populations, and ELF-based Lewis-like structures together with natural atomic charges of tropone **1**, and Cp **3**. The RED, in green, and the valence basin populations and natural atomic charges are provided in an average number of electrons, e. Negative charges are colored in red and positive charges in blue.

The ring electron density [54] (RED) of tropone **1**, i.e., the sum of the electron density populations of all V(Ci,Cj) disynaptic basins of the tropone ring core, 18.89 e, has a very high value compared to that of benzene, 16.62 e, which is the reference for aromatic compounds [54].

The ELF of Cp **3** shows the presence of two disynaptic basins, V(C9,C10) and V'(C9,C10), integrating a total of 3.40 e, associated with a C9–C10 partial double bond, and one V(C10,C11) disynaptic basin, integrating a total of 2.20 e, associated with a C10–C11 single bond. The ELF of the C11–C12 bonding region of Cp **3** is symmetrical to the C9C10 one.

Thus, ELF reveals that the C2–C3–C4–C5–C6–C7 bonding region of tropone **1** and the C9–C10–C11–C12 one of Cp **3** are similar to an unsaturated conjugated system.

Finally, the natural atomic charges of tropone **1** and Cp **3**, obtained by NPA [48,49], were studied (see Figure 1). With the exception of the C1 carbon of tropone **1**, which presents a high positive charge, +0.46 e, due to the strong polarization of the carbonyl C–O bonding region, the other six carbons of tropone **1** are negatively charged between −0.18 and −0.21 e because of the more electronegative character of the carbons than the hydrogens bound to them. The carbonyl O8 oxygen atom is strongly negatively charged, −0.58 e. On the order hand, the C9 and C10 carbons of Cp **3** are negatively charged by −0.18 and −0.23 e, respectively.

### 3.2. Analysis of the CDFT Reactivity Indices of the Reagents

Many studies devoted to polar reactions have shown that the analysis of the reactivity indices defined within the CDFT [18,19] is a powerful tool for understanding the reactivity

in polar reactions. The CDFT indices were calculated at the B3LYP/6-31G(d) computational level since it was used to establish electrophilicity [19] and nucleophilicity [19] scales. The global reactivity indices, namely, the electronic chemical potential μ, chemical hardness η, electrophilicity ω, and nucleophilicity *N*, of tropone **1** and Cp **3** are gathered in Table 1.

**Table 1.** B3LYP/6-31G(d) electronic chemical potential μ, chemical hardness η, electrophilicity ω, and nucleophilicity *N*, in eV, of tropone **1** and Cp **3**.

|  | μ | η | ω | *N* |
|---|---|---|---|---|
| tropone **1** | −4.28 | 4.20 | 2.18 | 2.75 |
| Cp **3** | −3.01 | 5.48 | 0.83 | 3.37 |

The electronic chemical potential [55] μ of Cp **3**, −3.01 eV, is above that of tropone **1**, μ = −4.28 eV (see Table 1). Consequently, along a polar reaction, the flux of the electron density will take place from Cp **3** to tropone **1**, these cycloaddition reactions being classified as of REDF [24].

The electrophilicity ω index [56] of tropone **1** is 2.18 eV, and it is classified as a strong electrophile within the electrophilicity scale [19]. In turn, the nucleophilicity *N* index [57] of tropone **1** is 2.75 eV, and it is classified as a moderate nucleophile within the nucleophilicity scale [19]. In contrast, its high nucleophilicity *N* index, *N* = 3.37 eV, classifies Cp **3** as a strong nucleophile participating in polar cycloadditions. Consequently, tropone **1** will act as a strong electrophile and Cp **3** will act as a strong nucleophile in polar cycloadditions of REDF [24].

In a polar reaction involving non-symmetric species, the most favorable reaction path involves the two-center interaction between the most electrophilic and the most nucleophilic centers of the reagents. Many studies have shown that the analysis of the electrophilic $P_k^+$ and nucleophilic $P_k^-$ Parr functions [58], resulting from the excess of spin electron density gathered via the GEDT [47], is one of the most accurate and insightful tools for the study of the local reactivity in polar and ionic processes. Hence, the electrophilic $P_k^+$ Parr functions of tropone **1** and the nucleophilic $P_k^-$ Parr functions of Cp **3** are analyzed (see Figure 2).

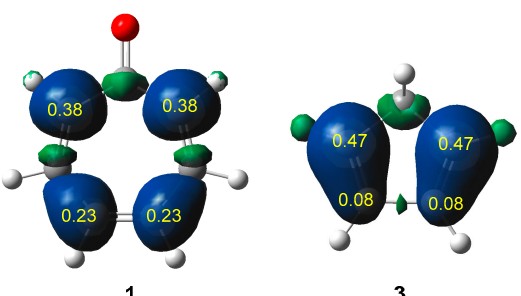

**Figure 2.** Three-dimensional representations of the Mulliken atomic spin densities of the radical anion of tropone **1** and the radical cation of Cp **3**, including the electrophilic $P_k^+$ Parr functions of tropone **1** and the nucleophilic $P_k^-$ Parr functions of Cp **3**.

The analysis of the electrophilic $P_k^+$ Parr functions at tropone **1** indicates that symmetric C2 and C7 carbons are the most electrophilic centers, $P_k^+$ = 0.38, followed by the symmetric C4 and C5 carbons, $P_k^+$ = 0.23. The electrophilic $P_k^+$ Parr functions on the C2 and C7 carbons are twice as high as those at the C4 and C5 carbons, indicating that the HOCA reaction of tropone **1** will be highly regioselective. On the other hand, the symmetric C9 and C12 carbons of Cp **3** are the most nucleophilic centers of this species, $P_k^-$ = 0.47 (see Figure 2).

Consequently, along the nucleophilic attack of Cp **3** to tropone **1**, the most favorable two-center interaction will take place between the C9 carbon of Cp **3** and the C2 carbon of tropone **1**.

It is worth emphasizing that the most electrophilic centers of tropone **1** are the symmetric C2 and C7 carbons, which are the two more negatively charged carbons, $-0.21$ e (see Figures 1 and 2). On the order hand, the positively charged carbonyl C1 carbon of tropone **1** is electrophilically deactivated, $P_k^+ = -0.05$. These findings support Domingo's proposal made in 2012 that the local electrophilic/nucleophilic behaviors of organic molecules are not simply caused by charges [59]. The most electrophilic center of a molecule is the one that accepts the highest amount of electron density resulting from the nucleophilic/electrophilic interactions taking place along the approach of the two reagents [59]. Interestingly, in some cases, such as in tropone **1**, the most electrophilic center is the one that is negatively charged [23,59].

### 3.3. Study of the Competitive Reaction Paths Associated with the Cycloaddition Reaction of Tropone **1** with Cp **3**

The cycloaddition reaction of tropone **1** with Cp **3** can take place along eight competitive reaction paths associated with the nucleophilic attack of the C9 carbon of Cp **3** on the electrophilic C2 or C4 carbons of tropone **1** (see Scheme 5). They are related to the formation of seven formal [2 + 4], [4 + 2], [6 + 4], or [8 + 2] CAs, showing the complexity of this reaction.

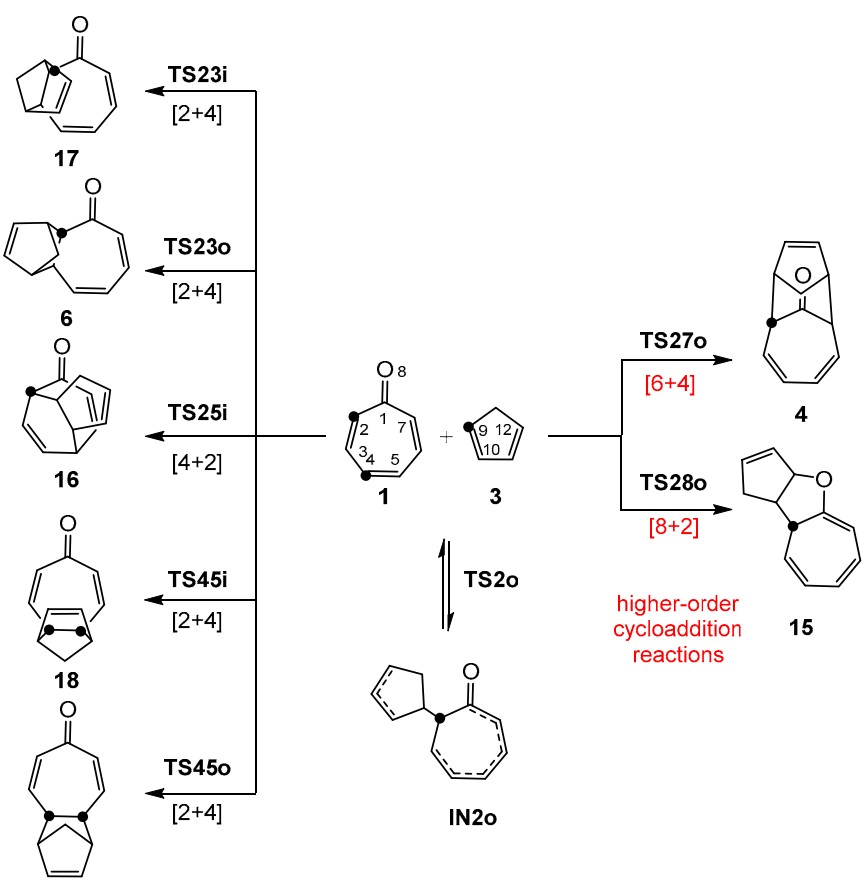

**Scheme 5.** Competitive reaction paths involved in the cycloaddition reaction of tropone **1** with Cp **3**.

Depending on the attack of Cp **3** to the C2 or C4 position of tropone **1,** the corresponding reaction paths take place through two different mechanisms: (i) Along the attack on the most electrophilic C2 position of tropone **1**, cycloaddition is characterized by a two-center interaction process involving a highly asynchronous TS; (ii) on the other hand, along the attack on the C4 position of tropone **1**, cycloaddition is characterized by a four-center interaction involving a synchronous TS. Along the two-center interaction processes, the C9–C10 double bond of Cp **3** can adopt three staggered conformations with respect to

the C2–C3 double bond of tropone **1**, yielding three isomeric constitutional CAs via three diastereoisomeric *pseudocyclic* TSs named *anti*, *gau1*, and *gau2* (see Scheme 6). In addition, Cp **3** can approach tropone **1** in two stereoisomeric modes named *in*, *i*, and *out*, *o*. In the *in* approach modes, the diene framework of Cp **3** situates over the triene framework of tropone **1**. TSs and CAs involved in this cycloaddition reaction have been named **TSxyi/o** and **CAxyi/o**, where **x** and **y** are the atomic positions of tropone **1** involved in the reaction, while **i** and **o** are related to the stereoisomeric *in* or *out* approach modes of Cp **3** to tropone **1**.

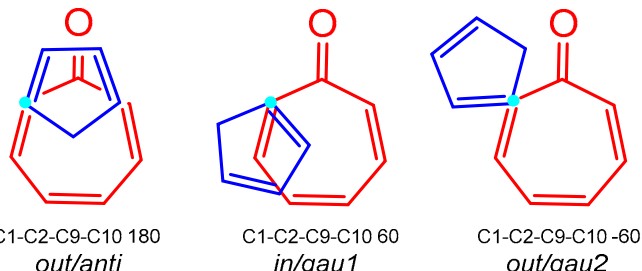

C1-C2-C9-C10 180    C1-C2-C9-C10 60    C1-C2-C9-C10 -60
*out/anti*      *in/gau1*      *out/gau2*

**Scheme 6.** Staggered conformational approach modes along the attack of Cp **3** on the C2 carbon of tropone **1**.

The analysis of the stationary points found along the eight reaction paths indicates that this cycloaddition reaction occurs via a one-step mechanism. The relative electronic energies, in the gas phase and benzene, are provided in Table 2, while the total electronic energies are shown in Table S2 in the Supplementary Materials.

**Table 2.** $\omega$B97X-D/6-311G (d,p) relative electronic energies, in kcal mol$^{-1}$, in gas phase, and benzene, of the stationary points involved in the cycloaddition reaction of tropone **1** with Cp **3**.

|  | **Gas Phase** | **Benzene** |  | **Gas Phase** | **Benzene** |
|---|---|---|---|---|---|
| **TS25i** | 17.0 | 17.4 | **16** | −29.4 | −28.3 |
| **TS27o** | 14.8 | 15.3 | **4** | −25.8 | −24.7 |
| **TS23i** | 16.7 | 16.9 | **17** | −16.4 | −15.3 |
| **TS23o** | 17.5 | 18.1 | **6** | −15.5 | −14.4 |
| **TS28o** | 25.1 | 25.5 | **15** | −21.5 | −19.9 |
| **TS2o** | 28.7 | 28.4 | **IN2o** | 28.6 | 27.9 |
| **TS45i** | 19.1 | 19.2 | **18** | −16.0 | −15.2 |
| **TS45o** | 19.3 | 19.5 | **19** | −14.9 | −14.0 |

The analysis of the relative energies provided in Table 2 indicates that the most favorable reaction path of this cycloaddition reaction is that associated with the nucleophilic attack of the C9 carbon of Cp **3** on the C2 position of tropone **1** via **TS27o**, 14.8 kcal·mol$^{-1}$, and the formation of the corresponding formal [6 + 4] **4** is exothermic by −25.8 kcal·mol$^{-1}$. Some interesting conclusions can be obtained from the analysis of the relative energies provided in Table 2: (i) An exhaustive analysis of the potential energy surface (PES) associated with this cycloaddition reaction shows its complexity; (ii) the most favorable **TS27o** is associated with the *out/anti* two-center interaction (see Scheme 6) between the more nucleophilic center of Cp **3**, the C9 carbon, and the most electrophilic center of tropone **1**, the C2 carbon, in clear agreement with the analysis of the Parr functions [58]; (iii) *out* **TS27o** corresponds with the *endo* approach mode of the diene framework of Cp **3** over the carbonyl C=O group of tropone **1**; (iv) this cycloaddition reaction is *out* stereoselective as **TS25i** is located 2.2 kcal·mol$^{-1}$ above **TS27o**. Note that **TS25i** and **TS27o** can be considered as a pair of *exo/endo* diastereomeric TSs yielding two constitutional isomeric CAs, **16** and **4**; (v) this cycloaddition reaction is *pseudocyclic* selective as *in/gau1* **TS23i** and *out/gau2* **TS2o** are found 1.9 and 13.9 kcal·mol$^{-1}$, respectively, above the most favorable *out/anti* **TS27o**. Note that *out/gau2* **TS2o** does not yield any CA in a single step but a very unstable intermediate **IN2o** (see Scheme 2 and Table 2); (vi) the *pseudocyclic in* **TS23i** is located only 0.3 kcal·mol$^{-1}$

below *in* **TS25i**; consequently, in this cycloaddition reaction, all competitive stereoisomeric reaction paths should be considered; (vii) this reaction is completely regioselective, as *in* **TS45i** associated with the [2 + 4] cycloaddition reaction involving the C4–C5 double bond of tropone **1** is located 4.3 kcal·mol$^{-1}$ above **TS27o**, which is in clear agreement with the analysis of the Parr functions; (viii) the formation of the formal [8 + 2] **15** is highly unfavorable as **TS28o** is 10.3 kcal·mol$^{-1}$ above **TS27o**; (ix) the analysis of the reaction energies associated with the formation of the seven CAs indicates that the formation of the experimental **4** is kinetically controlled. Note that the non-observed **16** is 3.6 kcal·mol$^{-1}$ more stable than **4**; finally, (x) the formal *endo* [6 + 4] CA **5**, first proposed by Ito [5] and Cookson [6] and further theoretically proposed by Houk [27], cannot be formed from the cycloaddition reaction of tropone **1** with Cp **3** (see later).

All stationary points involved in the cycloaddition reaction of tropone **1** with Cp **3** provided in Scheme 5 were also fully optimized at the B3LYP/6-311G (d,p) level. The total and relative energies are provided in Table S4 in Supplementary Materials. The B3LYP functional increases the activation energy of the most favorable reaction path via **TS27o** to 21.4 kcal·mol$^{-1}$ and decreases the exothermic character of the reaction to −5.1 kcal·mol$^{-1}$. Interestingly, this functional yields this cycloaddition reaction completely *out* stereoselective, [6 + 4] *pseudocyclic* selective, and regioselective as **TS25i**, **TS23i**, and **TS45o** are 2.6, 2.7, and 6.4 kcal·mol$^{-1}$ above **TS27o**, in complete agreement with the experimental outcomes [5,6]. Thus, although the B3LYP functional produces higher activation energies and lower reaction energies, it reproduces the relative electronic energies of the TSs in an improved manner, thus accounting for the only formation of the experimental formal *out* [6 + 4] CA **4**.

The inclusion of solvent effects of benzene in the optimizations does not produce any remarkable change in the relative energies. The relative energies of the TSs slightly increased by between 0.1 and 0.6 kcal·mol$^{-1}$, while those of CAs increased by between 0.8 and 1.6 kcal·mol$^{-1}$. This behavior results from the improved solvation of tropone **1** than the other species. In benzene, the *in* **TS27i** and **TS23i** remain 2.0 and 1.6 kcal·mol$^{-1}$, respectively, above the *out* **TS28o**.

The thermodynamic data of the stationary points involved in the four more favorable reaction paths associated with the HOCA reaction of tropone **1** with Cp **3** are provided in Table 3. Representations of the enthalpy and Gibbs-free energy profiles of these reaction paths are provided in Figure 3. The data for all stationary points involved in the eight studied reaction paths are provided in Table S1 in the Supplementary Materials.

**Table 3.** $\omega$B97X-D/6-311G(d,p) relative enthalpies ($\Delta$H in kcal·mol$^{-1}$), entropies ($\Delta$S in cal·mol$^{-1}$K$^{-1}$), and Gibbs-free energies ($\Delta$G in kcal·mol$^{-1}$), computed in benzene at 80 °C, of the stationary points involved in the four more favorable reaction paths associated with the cycloaddition reaction of tropone **1** with Cp **3**.

|  | $\Delta$H | $\Delta$S | $\Delta$G |
|---|---|---|---|
| **TS25i** | 18.0 | −48.4 | 35.1 |
| **TS27o** | 16.2 | −50.4 | 34.0 |
| **TS23i** | 17.7 | −47.1 | 34.3 |
| **TS23o** | 19.2 | −48.2 | 36.3 |
| **16** | −24.9 | −52.1 | −6.6 |
| **4** | −21.4 | −54.4 | −2.2 |
| **17** | −12.0 | −51.2 | 6.1 |
| **6** | −11.1 | −51.4 | 7.0 |

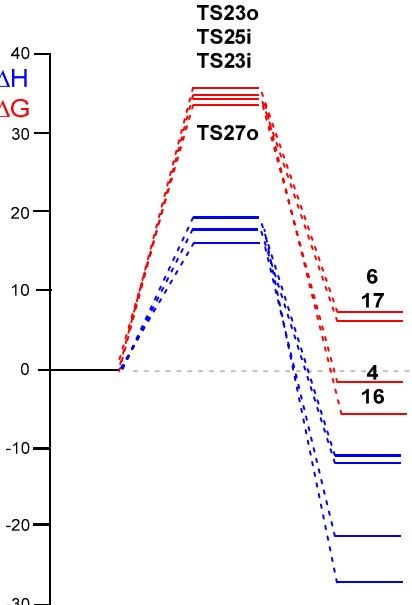

**Figure 3.** $\omega$B97X-D/6-311G (d,p) enthalpy ($\Delta$H in kcal mol$^{-1}$), in blue, and Gibbs-free energy ($\Delta$G in kcal mol$^{-1}$), in red, profiles, in benzene at 80 °C, of the stationary points involved in four more favorable reaction paths associated with the cycloaddition reaction of tropone **1** with Cp **3**.

The inclusion of the thermal corrections to the electronic energies increases the relative enthalpies by between 0.8 and 3.4 kcal·mol$^{-1}$; those of the TSs increased by less than 1.1 kcal·mol$^{-1}$. Similar selectivities were found by analyzing the activation enthalpies; thus, this HOCA reaction is *out* selective by 1.8 kcal·mol$^{-1}$. Including entropy and temperature terms to enthalpies increases the relative Gibbs-free energies by between 16.6 and 19.2 kcal·mol$^{-1}$ as a consequence of the unfavorable entropies associated with the bimolecular nature of this reaction: $\Delta$S ranges between −47.1 and −54.4 cal·mol$^{-1}$K$^{-1}$. The activation Gibbs-free energy associated with the formation of formal [6 + 4] CA **4** in this HOCA reaction rises to 34.0 kcal·mol$^{-1}$, and the reaction is exergonic by −2.2 kcal·mol$^{-1}$.

While the activation enthalpies provide selectivities similar to those obtained from the activation energies, the activation Gibbs-free energies clearly reduce them. Thus, while the *out* selectivity is lowered to 1.1 kcal·mol$^{-1}$, the *pseudocyclic* selectivity yielding the *in* [4 + 2] CA **18** diminished to 0.3 kcal·mol$^{-1}$. Thus, a mixture of the *out* [6 + 4] CA **4** and the *in* [4 + 2] CA **17** will be kinetically expected at this computational level.

Three interesting conclusions can be obtained by comparing the present MEDT study with the computational study recently reported by Houk: [27] (a) the ambimodal reactions proposed by Houk do not exist [60]. Each one of the feasible formal [m,n] CAs is obtained through a characterized reaction path associated with a single **TSxyi/o** (see Scheme 5); (b) the formal *in* [6 + 4] CA **5** originally proposed by Ito [5] and Cookson [6] can never be obtained from a direct formal [6 + 4] cycloaddition reaction of tropone **1** with Cp **3** as Houk proposed (see Scheme 4). In fact, the IRC from **TS-B** to the product reported by Houk connects with the formal [4 + 2] CA **16** and not with the formal *in* [6 + 4] CA **5** (see Scheme 5); finally, (c) Cookson reported that heating CA **4** in xylene reflux caused a disassociation to the original components tropone **1** and Cp **3** [6]. The formal [4 + 2] CA **16** is 4.4 kcal·mol$^{-1}$ more stable than the *out* [6 + 4] CA **4** (see Table 3 and Figure 3), but it was never experimentally observed. Consequently, the conversion of CA **4** into CA **15** via a Claisen rearrangement and CA **5** into CA **16** via Cope rearrangement that Houk suggested cannot experimentally take place (see Scheme 4) [27].

The geometries of the TSs are provided in Figure 4. While **TS45i** and **TS45o** are associated with slightly asynchronous and synchronous C–C single bond formation processes, respectively, as a consequence of equal C4 and C5 electrophilic activation, the other five TSs correspond to highly asynchronous C–C single bond formation processes initialized

by the nucleophilic attack of the C9 carbon of Cp **3** on the C2 carbon of tropone **1**. At these asynchronous TSs, the distances between the interacting carbons are found in the range from 2.023 (**TS23o**) to 1.828 (**TS28o**) Å, while the lengths between the non-interacting carbons range from 2.343 (**TS23o**) to 2.910 (**TS25i**) Å. Note that at the most unfavorable **TS2o**, the C2–C9 distance is very short, 1.741 Å, resulting from its very advanced character.

*a) TSs associated with a two-center interaction process*

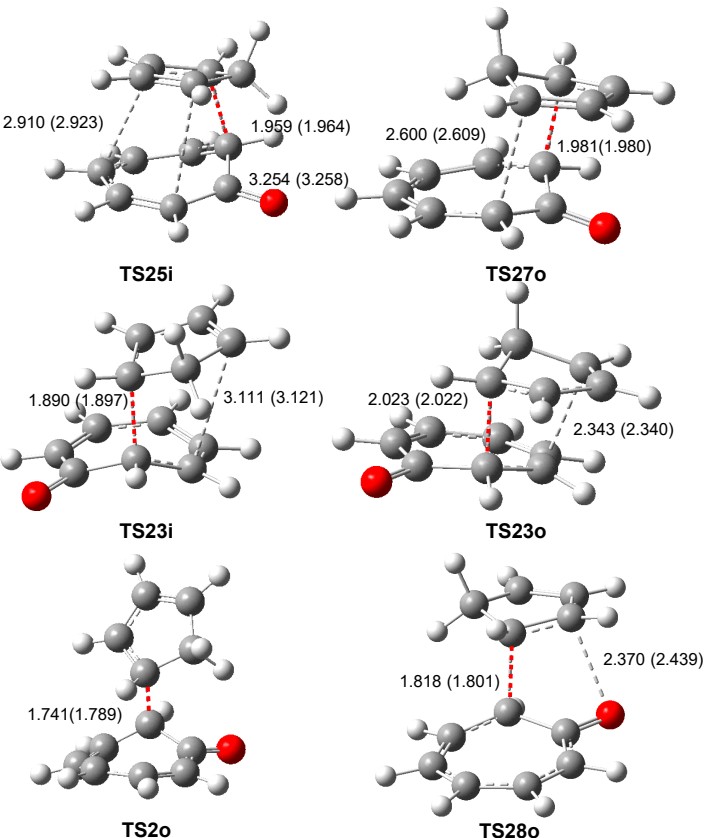

*b) TSs associated with a four-center interaction process*

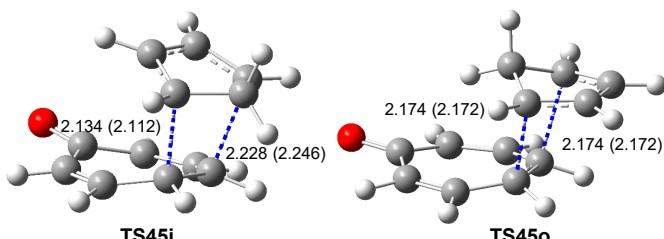

**Figure 4.** $\omega$B97X-D/6-311G(d,p) geometries of the TSs involved in the cycloaddition reaction of tropone **1** with Cp **3**. Distances are given in angstroms. The two-center interactions at the TSs are remarked in red, while the four-center interactions are in blue. Lengths in benzene are given in parentheses.

Interestingly, *pseudocyclic* TSs **TS25i** and **TS27o** show a different degree of asynchronicity at $\Delta l$ = 0.95 and 0.62 Å, respectively. This remarkable change of asynchronicity in these TSs is a consequence of two opposite electronic factors; while the favorable electronic interaction between the Cp framework and the carbonyl O8 oxygen brings them closer in *out* **TS27o**, the unfavorable electronic interaction between the Cp framework and the unsaturated ring of tropone **1** in *in* **TS25i** pushes them farther apart.

The analysis of GEDT at the TSs associated with the formation of the seven CAs permits assessing the polar character of this cycloaddition reaction [47]. GEDT values lower

than 0.05 e correspond to non-polar processes, while values higher than 0.20 e correspond to high polar processes. The GEDT values at the TSs are 0.20 e at **TS25i**, 0.19 e at **TS27o**, 0.18 e at **TS23i**, 0.18 e at **TS23o**, 0.27 e at **TS28o**, 0.14 e at **TS45i**, and 0.15 e at **TS45o**. The GEDT at the most favorable **TS27o**, 0.19 e, yielding the experimental [6 + 4] CA **4**, indicates that this HOCA reaction has a polar character, with the electron density fluxing from Cp **3** to tropone **1**; the reaction is, therefore, classified as REDF [24]. The high GEDT calculated at the most disfavored **TS28o**, 0.27 e, is a consequence of the very advanced character of this species; the C2–C9 distance at **TS28o** is 1.818 Å. Note that the GEDT at the TSs depends on the electrophilic and nucleophilic character of the two interacting molecules and the distance between them; the more advanced the TS, the higher the GEDT.

*3.4. BET Analysis of the Most Favorable Reaction Path Associated with the HOCA Reaction of Tropone* **1** *with Cp* **3**

To characterize the molecular mechanism of the HOCA reaction between tropone **1** with Cp **3**, a BET analysis [61] along the most favorable *out* [6 + 4] reaction path yielding the formal [6 + 4] CA **4** was performed. This methodology comprehensively describes molecular mechanisms by topologically scanning the bonding changes along the corresponding reaction path. BET analysis is provided in Section S2 in the Supplementary Materials. The main conclusions obtained from this BET analysis are herein summarized as follows:

(i) The bonding changes along the most favorable [6 + 4] reaction path are clearly asymmetric despite the symmetry of the reagents. This is because of the larger and, thus, more favorable electron density delocalization in the asynchronous process than in a hypothetical synchronous one [62];

(ii) Besides the asymmetry in the electron density reorganization, bond ruptures, and bond formations do not take place simultaneously but sequentially, emphasizing the non-concerted nature of this one-step HOCA reaction. Note that the formation of *pseudoradical* centers [63] required for the initial C2–C9 single bond formation demands the previous rupture of the C–C double bonds [47];

(iii) The formation of the two new C2–C9 and C7–C12 single bonds take place at C–C distances of ca. 1.93 and 2.13 Å, with initial populations of 0.95 and 1.04 e, respectively, through the C-to-C coupling of the corresponding C2/C9 and C7/C12 pairs of *pseudoradical* centers [47];

(iv) The formation of the first C2–C9 involves the most electrophilic center of tropone **1**, the C2 carbon, and the most nucleophilic center of Cp **3**, the C9 carbon, in agreement with the analysis of the Parr functions (see Section 3.2);

(v) The first C2 *pseudoradical* center appears at the most electrophilic center of tropone **1**, but the C9 *pseudoradical* center participates in a larger extension, 0.50 e, than the C2 one, 0.41 e, in the formation of the C2–C9 single bond;

(vi) The formation of the second C7–C12 single bond starts when the first C2–C9 single bond has reached 92% of its final population. The value for the synchronicity of this reaction, computed as the relative separation between the **S8** and **S14** IRC structures from the TS, is 0.08. Note that a value of 1.00 means a synchronous process. Consequently, the mechanism of the HOCA reaction between tropone **1** and Cp **3** is characterized as a non-concerted *two-stage one-step* mechanism [25];

(vii) The tropone carbonyl C–O group does not participate in the reaction, contrary to what could be expected by means of an extended conjugation with the carbonyl group.

Both geometrical and vibrational analyses of the most favorable **TS27o** and BET analysis of the bonding changes along the IRC show the non-concerted nature of the formation of the two C–C single bonds along this polar HOCA reaction and the symmetry of the two reagents, which is a behavior observed by Yamabe in 2002 who suggested that "*in spite of the symmetry of the reagents, the reaction proceeds* via *asymmetric TS that does not follow the W-H SR*" [9].

This behavior is a consequence of the fact that, when it is feasible, cycloaddition reactions prefer an asynchronous mechanism [62]. Thus, while the non-polar DA reaction

between butadiene and ethylene preferring the non-concerted synchronous mechanism as the stepwise mechanism via a diradical intermediate is 5.1 kcal·mol$^{-1}$ more unfavorable [64], the non-polar DA reaction between Cp **3** and styrene prefers a one-step mechanism via a highly asynchronous TS as the latter favors the asynchronous formation of the two *pseudoradical* centers involved in C–C single bond formations.

### 3.5. What Is the Origin of the Pseudocyclic Selectivity in this [6 + 4] HOCA Reaction?

The most favorable *out* **TS27o** yielding the formal [6 + 4] CA **4** has an *endo* stereoisomeric relationship between the conjugated system of Cp **3** and the carbonyl O8 oxygen of tropone **1**. *Endo* stereoselectivity has been explained within the FMO theory [16] by SOI present at the *endo* TSs, a concept introduced in 1983 by Gleiter and Bohm to explain the regio- and stereoselectivity in Diels–Alder reactions [28]. Today, SOI continues to be used to explain *endo* stereoselectivity in cycloaddition reactions. Thus, Houk suggested that the different SOI interactions at **TS-A** and **TS-B** were responsible for the only formation of *endo* [6 + 4] CA **4** (see Scheme 4) [27]. In 1999, Domingo proposed that the favorable electrostatic interactions existing in the *endo* zwitterionic TSs are responsible for the *endo* selectivity in polar Diels–Alder reactions [65]. In 2000, Salvatella et al. proposed that a combination of well-known mechanisms, such as solvent effects, steric interactions, hydrogen bonds (HBs), electrostatic forces, and others, can be invoked instead of SOI in the *endo/exo* selectivity of Diels–Alder reactions [66]. The *out* **TS27o** shows the minor asynchronicity of all asynchronous TSs, Δl = 0.62 Å. This low asynchronicity can be a consequence of the favorable electronic interactions between the C10H–C12H hydrogens of Cp **3**, being positively charged by +0.20 e each, and the carbonyl O8 oxygen, which is negatively charged by −0.58 e (see Figure 1), which approach each other. Figure 5 shows the maps of the molecular electrostatic potential (MEP) of *out* **TS27o** and its stereoisomeric *in* **TS25i**; note that **TS27o** and **TS25i** have a stereoisomeric *endo* and *exo* relationship. As shown in the map of the MEP of **TS27o**, the positively charged hydrogens of Cp **3** (blue region) are positioned over the carbonyl O8 oxygen, which is negatively charged (red region). Note that in *in* **TS25i**, these hydrogens are in the opposite direction relative to the carbonyl O8 oxygen. These favorable electrostatic interactions at **TS27o** are similar to those at *endo* zwitterionic TSs involved in polar Diels–Alder reactions [65]. These favorable electronic interactions appearing in polar cycloaddition reactions, which increase with the polar character of the cycloaddition reaction [65], are not possible in the other stereoisomeric TSs, as the carbonyl O8 oxygen is far from the Cp framework, thus presenting higher energies.

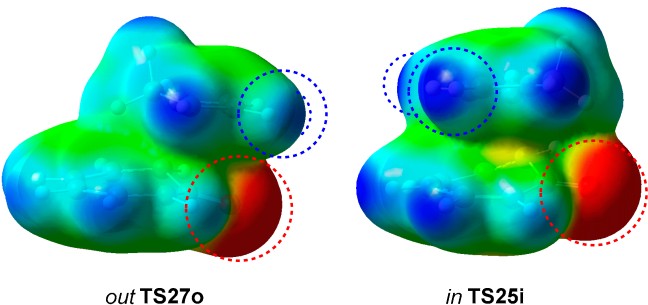

*out* **TS27o**          *in* **TS25i**

**Figure 5.** ωB97X-D/6-311G (d,p) Maps of the MEP for *out* **TS27o** and *in* **TS25i**. C10H–C12H hydrogens are highlighted by line-dashed blue circles, while the O8 oxygen is highlighted by a line-dashed red circle.

### 3.6. Why Is It Not Possible to Form the Formal in [6 + 4] CA 5 in the HOCA Reaction of Tropone 1 with Cp 3?

**TS25i** and **TS27o** are a pair of *in/out* diastereomeric TSs associated with the attack of the C9 carbon of Cp **3** on the most electrophilic C2 carbon of tropone **1**, yielding two constitutional isomeric formals [4 + 2] **16** and [6 + 4] **4** CAs. Figure 6 shows the *in/out* relationship between these TSs. Interestingly, as observed, the geometries of these TSs are

mainly resolute from the *gau1* and *anti* conformations adopted by the two frameworks along the C2–C9 single bond formation and not from any SOIs as Houk proposed [27]. Thus, while at the *anti/out* **TS27o**, the C12 carbon of Cp **3** is positioned 2.60 Å above the C7 carbon of tropone **1**; at *gau1/in* **TS25i** the C12 carbon of Cp **3** is positioned 3.25 Å above the C7 carbon, while the C10 carbon of Cp **3** is located 2.91 Å above the C5 carbon of tropone **1** (see the geometry of **TS25i** in Figure 4). Consequently, the IRC from *gau1/in* **TS25i** to products shows that, after passing this TS, the subsequent ring closure yields the formal [4 + 2] CA **16**, avoiding the formation of the expected formal [6 + 4] **5** as proposed by Houk (see Scheme 4 and the IRC in Figure S4 in Supplementary Materials)

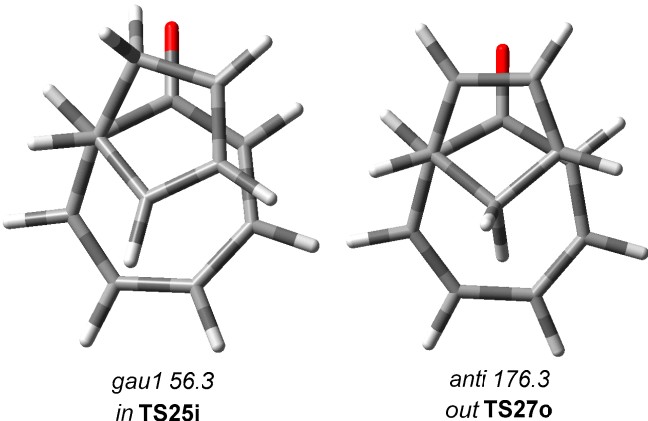

gau1 56.3
in **TS25i**

anti 176.3
out **TS27o**

**Figure 6.** View of **TS25i** and **TS27o** along the C2–C9 single bond formation. The C10–C9–C2–C1 dihedral angles are provided in sexagesimal degrees.

Figure 7 shows the ELF basin attractor positions of the stereoisomeric *pseudocyclic* **TS25i** and **TS27o** involved in the formation of constitutional isomeric formal [4 + 2] CA **16** and the formal [6 + 4] CA **4**. As observed, there is a complete similitude between the electronic structure of the two diastereoisomeric TSs. The ELF of the two TSs shows the presence of two V(C2) and V(C9) monosynaptic basins with a similar population, which are required for the subsequent formation of the C2–C9 single bond. These TSs are associated only with the C–C two-center interaction between the C2 and C9 carbons. The formations of the formal [4 + 2] or [6 + 4] CAs **16** and **4** are explicitly resolute after passing the corresponding TSs.

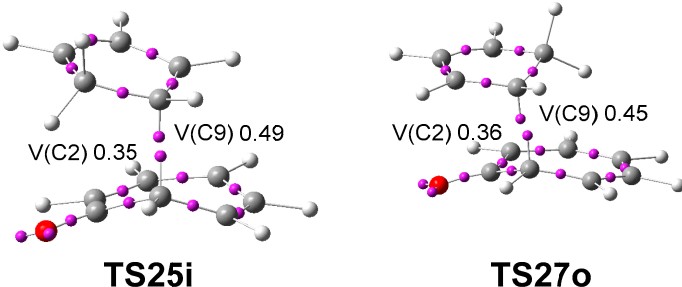

**TS25i**

**TS27o**

**Figure 7.** $\omega$B97X-D/6-311G (d,p) ELF basin attractor positions and basin populations, in average number of electrons, e, of the *pseudocyclic* **TS25i** and **TS27o** involved in the nucleophilic attack of the C9 carbon of Cp **3** on the C2 carbon of tropone **1** along the C2–C9 single bond formation.

## 4. Conclusions

The HOCA reaction of tropone **1** with Cp **3** yielding the *out* [6 + 4] CA **4**, experimentally reported by Ito [5] and Cookson [6] separately, has been studied within MEDT using DFT calculations at the $\omega$B97X-D/6-311G (d,p) computational level.

The ELF analysis of the electronic structure of tropone **1** and Cp **3** characterizes the structural behaviors of the two conjugated unsaturated systems. The C1–O8 bonding

region of tropone **1** is strongly polarized towards the electronegative O8 oxygen, causing the C1 carbon of tropone **1** to be highly positively charged by +0.45 e. The analysis of the CDFT reactivity indices of the reagents characterizes tropone **1** as a strong electrophile and Cp **3** as a strong nucleophile participating in the polar cycloaddition reactions of REDF. The analysis of the electrophilic Parr functions of tropone **1** indicates that the negatively charged C2 carbon is twice as electrophilically activated as the C4 one, while the positively charged C1 carbon is electrophilically deactivated.

A careful analysis of the PES associated with the reaction between tropone **1** and Cp **3** shows the complexity of this HOCA reaction for characterizing at least eight competitive reaction paths resulting in the nucleophilic attack of the C9 carbon of Cp **3** on the electrophilically activated C2 and C4 positions of tropone **1** in several stereoisomeric reaction paths. The most favorable reaction path allowing the formation of the formal *out* [6 + 4] CA **4** presents an activation enthalpy of 16.2 kcal·mol$^{-1}$, via **TS27o**, the reaction being exothermic by $-21.4$ kcal·mol$^{-1}$. The analysis of the relative enthalpies involved in this HOCA reaction indicates that it is highly stereo-, *pseudocyclic-*, and regioselective, explaining the formation of experimental formal *out* [6 + 4] CA **4** exclusively.

While the most favorable nucleophilic attack of Cp **3** on the most electrophilic C2 position of tropone **1** accounts for the regioselectivity of this HOCA reaction, the favorable electronic interactions present between the Cp framework and the negatively charged carbonyl O8 oxygen of tropone **1** at the most favorable *endo* **TS27o** and not SOI interactions as was suggested [27], which accounts for its stereo- and *pseudocyclic* selectivity. The GEDT value found at the most favorable **TS27o**, 0.19 e, quantifies the polar character of this HOCA reaction of REDF.

Interestingly, in spite of the symmetry of the two reagents, this HOCA reaction takes place via a highly asynchronous **TS27o** as a consequence of the most favorable two-center interactions taking place between the electrophilic C2 and nucleophilic C9 centers, a behavior already characterized in 2002 by Yamabe et al. [9]. This behavior goes against the concept of "concerted" proposed by Lewis in 1929 [67], and it is further used by Woodward and Hoffmann to define "pericyclic reactions" [15]. When feasible, cycloaddition reactions prefer a non-synchronous C–C single bond formation as in this polar HOCA reaction.

Despite the reaction path associated with the formation of the experimental formal *out* [6 + 4], CA **4** is the least asynchronous one among those involving the attack on the C2 carbon of tropone **1**, and the BET analysis of this HOCA reaction indicates that it takes place through a non-concerted *two-stage one-step* mechanism [25], in which the formation of the second C–C single bond starts when the first C–C single bond has reached 92% of its final population.

The present MEDT study of the HOCA reaction of tropone **1** with Cp **3** contrasts with the recent computational study based on Houk's model of "pericyclic ambimodal reactions" [27,60]. HOCA reactions are not "pericyclic reactions" [23,34], and the "ambimodal reaction" concept has no chemical meaning as every reaction path found in this cycloaddition reaction undoubtedly connects each one of the eight characterized TSs with the corresponding formal [m,n] Cas.

**Supplementary Materials:** The following supporting information can be downloaded at: https://www.mdpi.com/article/10.3390/chemistry4030052/s1, BET analysis of the most favourable reaction path associated with the HOCA reaction of tropone **1** with Cp **3**. Figure with the gas-phase B3LYP/6-311G (d,p) geometries of the TSs involved in the HOCA reaction of tropone **1** with Cp **3**. Figure with the $\omega$B97X-D/6-311G (d,p) geometries of **TS2o** and **IN2o**. Figure the IRC connecting tropone **1** and Cp **3** with the formal [4 + 2] CA **16**. Table with the $\omega$B97X-D/6-311G (d,p) total electronic energies, in gas phase and benzene, of the stationary points involved in the HOCA reaction of tropone **1** with Cp **3**. Table with the B3LYP/6-311G (d,p) gas phase total and relative energies of the stationary points involved in the HOCA reaction of tropone **1** with Cp **3**. $\omega$B97X-D/6-311G (d,p) thermodynamic data of the stationary points involved in the HOCA reaction of tropone **1** with Cp **3**.

**Author Contributions:** Conceptualization, L.R.D., M.R.-G. and P.P.; methodology, L.R.D., M.R.-G. and P.P.; investigation, L.R.D., M.R.-G. and P.P.; resources, L.R.D., M.R.-G. and P.P.; writing—original draft preparation, L.R.D.; writing—review and editing, L.R.D., M.R.-G. and P.P.; supervision, L.R.D.; project administration, L.R.D., M.R.-G. and P.P.; funding acquisition, L.R.D., M.R.-G. and P.P. All authors have read and agreed to the published version of the manuscript.

**Funding:** This research was funded by the Spain Ministry of Science and Innovation (MICINN) (LRD and MRG), project PID2019-110776GB-I00 (AEI/FEDER and UE), by FONDECYT/ANID–Chile through Project No. 1221383 (PP), and by the European Union's Horizon 2020 research and innovation program under the Marie Skłodowska-Curie grant agreement No. 846181 (MRG).

**Institutional Review Board Statement:** Not applicable.

**Informed Consent Statement:** Not applicable.

**Data Availability Statement:** Not applicable.

**Acknowledgments:** This work has been supported by the Ministry of Science and Innovation (MICINN) of the Spanish Government, by FONDECYT/ANID–Chile, and by the European Union's Horizon 2020 research and innovation programme under the Marie Skłodowska-Curie grant. L.R.D. also acknowledges Fondecyt Cooperación Internacional for its continuous support.

**Conflicts of Interest:** The authors declare no conflict of interest.

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
