# Peer review of "Unveiling the Chemistry of Higher-Order Cycloaddition Reactions within the Molecular Electron Density Theory"

_chemistry, doi:10.3390/chemistry4030052_

Round 1

Reviewer 1 Report

The article is devoted to the higher-order cycloaddition (HOCA) reaction of tropone with cyclopentadiene (Cp) has been studied within Molecular Electron Density Theory. As a result of the theoretical study, eight competing reaction routes were identified. The authors of the article identified and characterized the most favorable reaction route. The theoretical study carried out confirms the experimentally observed high stereo-, pseudocyclic- and regio-selectivities. The theoretical study of the cycloaddition reaction was carried out on the basis of methods well known in the literature. Thus, the authors do not propose new theoretical approaches for studying the described reaction. This article by Domingo et al. is based on previous published theoretical studies by the authors. However, this article is of interest and may be published. It follows from the study that within MEDT using DFT calculations at the B97X-D/6-311G(d,p) computational level is an effective method for studying the tropone with cyclopentadiene cycloaddition reaction. This is confirmed by experimentally observed facts. With the help of the calculations carried out in the article, a detailed analysis of the mechanism of the reaction under study is given, and this is an undoubted advantage of the work, which is worthy of publication.

Author Response

We thanks the nice comments made by Reviewer 1, which accepts the manuscript publication.

Reviewer 2 Report

This manuscript presents an investigation into the product distribution from a Diels-Alder style reaction between tropone and cyclopentadiene. This is seemingly in response to or extension of a subset of the reactions investigated in the recently published work from Jamieson et al., JACS, 2021. The primary goal of this work appears to be to make a distinction for the driving forces in the reactivity of this one system, this to provide presumably deeper insight into the true nature of the reactivity as described by select Density Functional Theory calculations. To this end, the authors provide a series of additional calculations and analyses that they have generally gathered under the umbrella of Molecular Electron Density Theory.

Much of the work presented in this manuscript is of relatively high quality and builds a case for the claims that the authors are making. Some of the points require subtle distinctions, making the associated assertions weaker than one would hope, but the aggregate push is mostly clear. In general, the authors would be better served with a less argumentative tone towards Houk et al. With it, the authors come across as bitter or petty rather than scientifically unbiased. This work is likely suitable for publication in the MDPI Chemistry journal following minor refinements as decided upon and implemented by the editor and authors. Some suggestions follow.

- In principle, this work is not really a review but a targeted investigation into this specific system and a critical commentary on one part the previously referred to publication. As such, the motivating systems described in Schemes 2 and 3 are not very relevant. Their inclusion pads this work, but does not serve to enhance the messaging, making it potentially detrimental to the authors' efforts. Losing the descriptions here could help this work. These could be summarized in a single sentence that states something along the lines of "Tropone is a reactive partner of interest in a variety of cycloaddition reactions [7-9]."

- Lines 213 & 234: Why the separate paragraph from the one previous? There seems to be a rather uneven presentation with a lot of these single sentence paragraphs. 

- Lines 245-246: What is the point of Ref. 22 here? It is just a general study of non-covalent interactions and does not include subtle cases on where Frontier orbital theory works despite potentially unfavorable electrostatic interactions. This paragraph overall is a somewhat weakly supported assertion. Frontier orbital theory covers this case and it is unnecessary to invoke a self-citation as some sort of profound counterpoint. This is one of the minor/subtle distinctions that fails to bolster the authors' case. Its inclusion seems to simply make a point that electrostatic non-covalent interactions are important and discriminating in some cases also not important and not discriminating in those same cases. The muddled messaging through cherry picking of projected charge at atomic centers is unhelpful.

-  DFT is notoriously poor at handling non-covalent interactions such as dispersion. The authors address this concern with a both technically and grammatically incorrect final sentence. This is a 6+ line monstrosity that is flawed and leaves a very bad look at the end of this work. For example, the default B3LYP hybrid functional does not handle well such non-covalent interactions without applied corrections. This entire conclusion is somewhat stapled together with a slew of single sentence paragraphs. Please clean up this presentation. If much of the case that the authors are presenting depends on electrostatic non-covalent interactions, this concern with chosen model theory should be addressed right up front when discussing the computational methods.

- For a work that is rather critical of published findings from other labs, it would very much help if the numbers presented were validated against previously published findings. This is unfortunately impossible given the different choice of basis functions used here versus Ref. 27. By not having directly comparable numerical basis, the authors are weakening their case. Instead of providing detailed specific findings, the presented results are comparable only in a relative sense, this with a Pople basis set that is generally considered inferior in accuracy to the calculations in publications over which the authors are highly critical. These authors are doing themselves no favors with their choices. They cast a cloud of uncertainty over the findings, with them possibly being interpreted as a numerical fluke in some cases. 

The authors should at the very least validate some of their choices with a couple calculations that reproduce exactly other results. This will provide some assurance that the subtle effects/differences are not simply numerical happenstance.

Author Response

- Lines 213 & 234: Why the separate paragraph from the one previous? There seems to be a rather uneven presentation with a lot of these single sentence paragraphs. 

R. In agreement with the referee’s suggestion, this paragraph has been joined with the previous one.

- Lines 245-246: What is the point of Ref. 22 here? It is just a general study of non-covalent interactions and does not include subtle cases on where Frontier orbital theory works despite potentially unfavorable electrostatic interactions. This paragraph overall is a somewhat weakly supported assertion. Frontier orbital theory covers this case and it is unnecessary to invoke a self-citation as some sort of profound counterpoint. This is one of the minor/subtle distinctions that fails to bolster the authors' case. Its inclusion seems to simply make a point that electrostatic non-covalent interactions are important and discriminating in some cases also not important and not discriminating in those same cases. The muddled messaging through cherry picking of projected charge at atomic centers is unhelpful.

R. We thank this referee’s comment. It was a mistake, as reference 22 should be reference 23. It has been corrected in this revised version.

-  DFT is notoriously poor at handling non-covalent interactions such as dispersion. The authors address this concern with a both technically and grammatically incorrect final sentence. This is a 6+ line monstrosity that is flawed and leaves a very bad look at the end of this work. For example, the default B3LYP hybrid functional does not handle well such non-covalent interactions without applied corrections. This entire conclusion is somewhat stapled together with a slew of single sentence paragraphs. Please clean up this presentation. If much of the case that the authors are presenting depends on electrostatic non-covalent interactions, this concern with chosen model theory should be addressed right up front when discussing the computational methods.

R. We agree with the referee’s comments. Accordingly, the mentioned paragraph has been removed in this revised version.

- For a work that is rather critical of published findings from other labs, it would very much help if the numbers presented were validated against previously published findings. This is unfortunately impossible given the different choice of basis functions used here versus Ref. 27. By not having directly comparable numerical basis, the authors are weakening their case. Instead of providing detailed specific findings, the presented results are comparable only in a relative sense, this with a Pople basis set that is generally considered inferior in accuracy to the calculations in publications over which the authors are highly critical. These authors are doing themselves no favors with their choices. They cast a cloud of uncertainty over the findings, with them possibly being interpreted as a numerical fluke in some cases. 

The authors should at the very least validate some of their choices with a couple calculations that reproduce exactly other results. This will provide some assurance that the subtle effects/differences are not simply numerical happenstance.

R. We thanks this referee’s comment. The choice of different sets of functionals and basis sets can give different energy results. It is the case of our calculations with respect to those of reference 27, but the end of this MEDT study is not to reproduce exactly any numeric result but to explain the single formation of the formal [6+4] cycloadduct 4 through this higher-order cycloaddition reaction. Unlike the recently published MEDT studies of tropone with ethylene derivatives, the participation of a diene system as cyclopentadiene notably increases the feasible competitive reaction paths in this higher-order cycloaddition reaction. All of them have been considered and analyzed in this theoretical study. The present MEDT provides a theoretical rationalization for the only formation of the formal [6+4] cycloadduct 4.

According to the referee’s comment, the last paragraph of the Conclusions part has been removed in this revised version.

Reviewer 3 Report

The manuscript by Domindo at al. Describes results of the computational study on cycloaddition reaction between tropone and cyclopentadiene. Cycloaddition reactions are important for the synthesis of various heterocycles, and, therefore, investigation of their mechanism is of interest. Results may be interesting for readers working in the area of cycloaddition reactions. This work may be accepted for publication in Chemistry but after revision in accord with comments below.

1. Discussion of the ELF analysis of tropone on page 4 is a copy-paste from a previous article published by the authors (Ref. 23). It is sufficient to give a reference to these published results and not to repeat them in each publication with tropone.

2. Discussion based on gas phase geometries and on the total energies has no meaning because it has no relation to the real systems. These data should be moved to Supporting Information, and all discussion should be based on Gibbs free energies in solution.

3. Table 3 shows that wB97X-D/6-311G(d,p) does not predict correctly experimental observations (experimental detection of only 4 but theoretical prediction of the formation of a mixture 4+17). Why not to use another, more appropriate functional which could reproduce better the experiment? The authors mentioned that B3LYP may be more appropriate.

4. Page 10, line 350. Figure 97 -> Figure 3.

Author Response

- Discussion of the ELF analysis of tropone on page 4 is a copy-paste from a previous article published by the authors (Ref. 23). It is sufficient to give a reference to these published results and not to repeat them in each publication with tropone.
R. We agree with the referee’s comment that ELF of tropone is given in reference 23; note that ELF of cyclopentadiene is also given in many manuscripts. However, it is important to keep this information in the text for easy comparison and visualization of its structure with respect to the diene derivatives.
- Discussion based on gas phase geometries and on the total energies has no meaning because it has no relation to the real systems. These data should be moved to Supporting Information, and all discussion should be based on Gibbs free energies in solution.
R. We agree with this comment, but we would like to remark that any theoretical study is based on models that may not relate to the real system. Some authors give the discussion part based only on the Gibbs free energies, see reference 27, but organic chemists are more amiable with enthalpies. The structure of the Discussion part is like others published by Domingo, see references 23 and 33, and therefore we believe that it should not be modified.
- Table 3 shows that wB97X-D/6-311G(d,p) does not predict correctly experimental observations (experimental detection of only 4 but theoretical prediction of the formation of a mixture 4+17). Why not to use another, more appropriate functional which could reproduce better the experiment? The authors mentioned that B3LYP may be more appropriate.
R. We completely agree with the referee’s comment. With our extensive experience in the field of the theoretical organic chemistry, we would like to comment that while DFT was an important advance in the study of chemical reactivity, the need to use functionals in the calculations was a huge misfortune to obtain accurate energies able to reproduce exactly the experimental outcomes. For example, many authors use the M062X functional in the study of organic reactivity; in our experience, it is one of the worst. Physicists have developed more than 1000 functional, showing this problem within DFT.
Fortunately, any DFT functional gives an accurate electron density distribution, the quantitative analysis of which is the purpose of our MEDT studies devoted to explain organic chemical reactivity.
- Page 10, line 350. Figure 97 -> Figure 3.
R. This mistake has been corrected in agreement with the referee’s suggestion.